# Leveraging Imitation Learning and LLMs for Efficient Hierarchical Reinforcement Learning

## Abstract

In this paper, we introduce an innovative framework that combines Hierarchical Reinforcement Learning (HRL) with Large Language Models (LLMs) to tackle the challenges of complex, sparse-reward environments. A key contribution of our approach is the emphasis on imitation learning during the early training stages, where the LLM plays a crucial role in guiding the agent by providing high-level decision-making strategies. This early-stage imitation learning significantly accelerates the agent's understanding of task structure, reducing the time needed to adapt to new environments. By leveraging the LLM's ability to generate abstract representations of the environment, the agent can efficiently explore potential strategies, even in tasks with high-dimensional state spaces and delayed rewards. Our method utilizes the LLM to assist action sampling via a dynamic annealing strategy and aids the policy learning process through an LLM-based policy and value regularizer. This approach reduces computational costs and enhances the agent's learning process. Experimental results across three environments—MiniGrid, NetHack, and Crafter—demonstrate that our method significantly outperforms baseline LLM-based HRL algorithms in terms of training speed and success rates. The imitation learning phase proves critical in enabling the agent to adapt quickly and perform efficiently, highlighting the potential of integrating LLMs into HRL for complex tasks.

## 1 Introduction

Reward sparsity is a persistent challenge in the early stages of exploration for reinforcement learning (RL) environments. As these environments grow more complex, the difficulty for agents to encounter rewards during initial exploration increases significantly. Researchers have continuously worked on addressing reward sparsity to efficiently train agents (Vecerik et al., 2017; Hare, 2019). Hierarchical reinforcement learning (HRL) offers a promising approach to addressing these issues. In HRL, the decision-making problem is decomposed into two levels: high-level decision making, referred to as an *option*, and low-level decision making, referred to as an *action*. Experimental evidence (Kulkarni et al., 2016; Nachum et al., 2018b) suggests that HRL can address challenges that traditional RL algorithms struggle with, demonstrating superior performance in general environments. However, despite HRL's potential, most approaches rely on predefined options and often require pre-training of the option networks. This makes HRL notoriously difficult to implement and tune effectively.

LLMs have shown potential in mitigating some of these challenges. Recently, LLMs have demonstrated their versatility across many domains, including natural language processing, code generation, and decision-making tasks such as game playing and intelligent question-answering (Du et al., 2023). A series of recent works (Zhou et al., 2024) suggest that LLMs can be leveraged as high-level decision-makers to enhance the performance of RL agents. While promising, it is well-known that LLM-based approaches are resource-intensive, which makes them less ideal compared to traditional RL methods that are computationally lighter. This leads to the following question:

*What is the most effective strategy to accelerate RL with LLMs?*

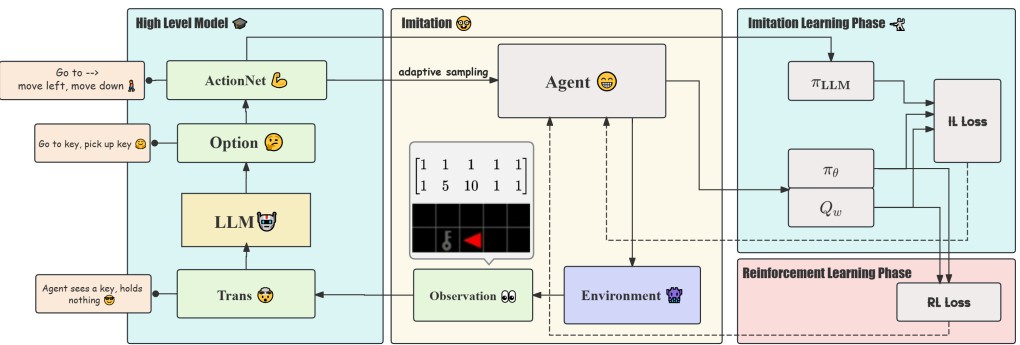

Figure 1: Algorithm framework of IHAC. Our framework is a two-phase algorithm designed to improve the learning efficiency of reinforcement learning agents by incorporating high-level guidance from LLM. In the first phase, imitation learning is used, where the agent benefits from LLM-generated options to accelerate its exploration and learning process. The second phase transitions to standard reinforcement learning, where the agent refines its policy and value networks, initially trained with the help of the LLM, to further optimize its decision-making abilities. By balancing imitation learning with reinforcement learning, IHAC achieves a more efficient learning process, especially in complex and long-term environments with sparse rewards.

To address this question, we propose the Imitation Hierarchical Actor-Critic (IHAC) framework, which accelerates the RL process in a hierarchical RL approach by incorporating high-level instructions from LLMs. The overview of IHAC is displayed in Figure 1. Our contributions are listed as follows.

- We propose a novel approach to imitation learning by framing the decision-making process as a hierarchical RL problem, where high-level options represent overarching instructions that an agent can follow to enhance decision-making. Our key contribution lies in the two-phase design of the proposed IHAC framework, which effectively balances imitation learning and reinforcement learning. In the first phase, IHAC leverages an external LLM to generate a higher-policy action distribution, which serves as input to both the actor and critic for simultaneous imitation learning. This ensures that the LLM's guidance is maximally utilized, enabling the agent to learn both action selection and value estimation early on, when its interaction with the environment is limited. In the second phase, IHAC transitions to a standard RL algorithm (e.g., PPO) to fine-tune the policy further. This design significantly accelerates learning while reducing reliance on LLM in later stages, achieving computational efficiency without sacrificing performance.

- To fully harness the power of the LLM, we introduce an adaptive sampling strategy that combines input from both the RL agent and the LLM to derive better actions during the imitation learning phase. Additionally, we propose an adaptive policy training strategy that facilitates a more precise approximation of the agent's policy with the help of the LLM. Both adaptive strategies help us balance the LLM's guidance with the RL agent's learning potential.

- For our experiments, we test our algorithm on several standard hierarchical RL benchmarks, such as MiniGrid Chevalier-Boisvert et al. (2023). Compared to existing baselines, our algorithm demonstrates superior performance and greater efficiency, particularly in terms of token usage, due to the incorporation of the LLM and our adaptive algorithm design.

## 2 RELATED WORKS

### 2.1 HIERARCHICAL REINFORCEMENT LEARNING (HRL)

The emergence of hierarchical reinforcement learning (HRL) has proven to be beneficial for addressing complex, large-scale problems and sparse, delayed rewards (Nachum et al., 2019). By introducing sub-goals and training multi-level structures, HRL effectively enhances exploration. However, a key challenge in HRL lies in establishing high-quality hierarchical structures to improve training

efficiency. One approach to establishing hierarchical structures involves manually setting them based on tasks, such as graph-guided reinforcement learning (Lee et al., 2022; Gieselmann & Pokorny, 2021), or by artificially setting sub-goals (Florensa et al., 2017; Tessler et al., 2017). The intervention of human priors renders the model non-generalizable. Additionally, there have been proposals to allow agents to autonomously learn sub-tasks. Some achieve this by associating the sub-task space with the current state (Zhang et al., 2020; Nachum et al., 2018a; Vezhnevets et al., 2017), while others consider predicting the next sub-goal during the training process (Pitis et al., 2020).

To enhance training efficiency, some researchers have tackled this issue by constraining the tasks of higher-level agents, limiting the state space of sub-tasks to the adjacent range of the current state (Zhang et al., 2020). Luo et al. (2023) proposed introducing attention rewards to enable higher-level agents to focus more on the environment. Others build a causality-driven hierarchical reinforcement learning framework, leveraging a causality-driven discovery instead of a randomness-driven exploration (Hu et al., 2023). *Hierarchical in-Context Reinforcement Learning* (HCRL) further integrates in-context learning with HRL frameworks to dynamically generate sub-goals based on ongoing task progress. This framework, through reflection and modularity, allows for correction of sub-task errors and improves task efficiency (Sun et al., 2024). *LLM Augmented Hierarchical Agents* have also leveraged LLMs to improve high-level decision-making, leading to better performance in long-horizon tasks (Prakash et al., 2024).

## 2.2 LLM AGENT

Previous research has demonstrated that language can facilitate the construction of abstract representations of both the environment and goals, especially in scenarios involving high-dimensional state spaces and long planning horizons (Lin et al., 2023; Andreas et al., 2017; Akakzia et al., 2020; Mirchandani et al., 2021). For example, Wu et al. (2019) showed that language could assist in learning complex tasks, even when naive goal representations fail. By employing language-guided hindsight goal relabeling, their approach achieved significant performance improvements. Jiang et al. (2019) further explored the use of language in goal specification, demonstrating that high-level policies could achieve their objectives by composing sub-policies guided by language. These studies underscore the utility of language as a compositional tool in RL, enabling more effective learning and generalization in complex, temporally-extended tasks.

Language-assisted reinforcement learning provides agents with a high-level understanding of tasks and environments. LLMs, with their powerful language processing capabilities, further expand this field by offering more nuanced task guidance, abstract representations, and decision support (Kwon et al., 2023; Du et al., 2023; Hu et al., 2023). Yao et al. (2023) introduced ReAct, a method where the LLM generates "thoughts" to address problems based on observations. Building on ReAct, Shinn et al. (2024) developed Reflexion, which leverages a few-shot verbal feedback approach to improve decision-making abilities. Ahn et al. (2022) provided another viewpoint, endowing LLMs with foundational "skills" along with corresponding value functions and affordance functions. With LLMs' assistance, each action selection is comprehensively considered, integrating the reward (value) of the current action and its feasibility (affordance). Nottingham et al. (2023) describe how to leverage Brief Language Inputs for Decision-making Responses (BLINDER) to condense the input text to LLMs, effectively reducing the cost of invoking LLMs by approximately sixfold. Building on these foundational studies, Li et al. (2023) proposed Interactive Task Planning (ITP), a framework that leverages LLMs for dynamic task planning and replanning in robotic systems. By integrating high-level planning with low-level skill execution, ITP enables robust adaptation to user feedback and novel tasks without the need for task-specific training or extensive prompt engineering.

## 2.3 IMITATION LEARNING

In many previous studies, imitation has become a strategy to develop the performance of reinforcement learning.

Introduced by Price & Boutilier (2003), the student agent can observe the state transitions induced by the mentor's actions and use the information gleaned from these observations to update the estimate of the value of its own states and actions, thus accelerating the process of RL training.

What's more, Oh et al. (2018) describes the Self-Imitation Learning strategy, which can utilize experience from the past. The agent will learn from its past options and rewards to develop itself, i.e., the agent itself is both the student and the mentor.

As for hierarchical cases, Le et al. (2018) provides a method to expand imitation learning to HRL. It provides a method to divide HRL into high-level IL and low-level RL. The meta-controller takes actions by imitating an expert agent, while the lower controller receives commands from the meta-controller, which is the same as in HRL.

Recent work in imitation learning has introduced the *Hindsight Modular Reflection* (HMR) framework, which allows agents to learn from both failed and successful trajectories by reflecting on sub-task failures and incorporating those insights into future decisions. This method effectively addresses sparse reward environments by using hindsight experience replay to convert hard-to-achieve goals into manageable intermediate sub-goals (Sun et al., 2024).

## 3 METHODOLOGY

### 3.1 PRELIMINARY: DEFINITION OF AGENTS

**Hierarchical Markov Decision Process** We define a Hierarchical Markov Decision Process (HMDP) as $\mathcal{M} = (\mathcal{S}, \mathcal{A}, \mathcal{O}, P, r, \gamma)$, where $\mathcal{S}$ is the state space, which includes all possible states the system can be in. $\mathcal{A}$ is the action space, which consists of all possible low-level actions an agent can take. $\mathcal{O}$ is the option space, which consists of all high-level actions. For instance, for the MiniGrid game, the high-level action can be some general instructions ("grab a bear"), and the low-level action can be the detailed action ("move right for 2 seconds"). $P : \mathcal{S} \times \mathcal{A} \times \mathcal{S} \rightarrow [0, 1]$ is the transition probability function, where $P(s'|s, a)$ represents the probability of transitioning from state $s \in \mathcal{S}$ to state $s' \in \mathcal{S}$ after taking action $a \in \mathcal{A}$. $r : \mathcal{S} \times \mathcal{A} \rightarrow \mathbb{R}$ is the reward function, which defines the reward $r(s, a)$ the agent receives after taking action $a$ in state $s$. $\gamma \in [0, 1]$ is the discount factor, which determines the importance of future rewards relative to immediate rewards. In our work, we are interested in finding a policy $\pi(\cdot|\cdot) : \mathcal{S} \rightarrow \Delta(\mathcal{A})$, which maximizes the expected cumulative reward $\mathbb{E}[\sum_{t=0}^{\infty} \gamma^t r_t]$.

**Large Language Model** Our framework utilizes a language model LLM to help the agent to find the optimal policy. We briefly introduce the details of them here. Let $\mathcal{L}$ be the language space that consists of sentences in the natural language. Let $\text{LLM}(\cdot|\cdot) : \mathcal{L} \rightarrow \Delta(\mathcal{O})$ be the LLM that takes the language as its input which outputs an option. For instance, the input of an LLM can be "Agent sees a key and hold nothing", and the output is some high-level option like "go to key, pick up key".

---

**Algorithm 1** IHAC

---

**Require:** Language model LLM, predefined action net `ActionNet`, translator `Trans`, imitation ratio $\lambda_t$, policy-value balance $\alpha$
1: $\mathcal{T} \leftarrow \{\}$, initialize $\pi_\theta$ and $Q_w$, $t \leftarrow 1$
2: **for** $i = 1, \ldots, n$ **do**                                  ▷ **Phase I**: imitation learning phase
3:      Receive initial state $s_1$, step $h = 1$, Done = False
4:      **while** not Done **do**
5:          Set $\pi_{\text{LLM}}(\cdot|s_h) \leftarrow \text{ActionNet}(\text{LLM}(\cdot|\text{Trans}(s_h)), s_h)$
6:          Sample action by $a_h \sim (1 - \lambda_t)\pi_\theta(\cdot|s_h) + \lambda_t \cdot \pi_{\text{LLM}}(\cdot|s_h)$
7:          Take action $a_h$, observe $r_h$, update state to $s_{h+1}$, update Done, $h \leftarrow h + 1$, $t \leftarrow t + 1$
8:      **end while**
9:      $\mathcal{T} \leftarrow \mathcal{T} \cup \tau = (s_1, a_1, r_1, \ldots, s_h, a_h, r_h)$, update $\pi_\theta, Q_w$ following equation 1
10: **end for**
11: Run standard RL algorithm (i.e., PPO) starting from $\pi_\theta$ and $Q_w$                ▷ **Phase II**: RL phase

---

### 3.2 PROPOSED ALGORITHM

We propose our algorithm, IHAC, as outlined in Algorithm 1 . Broadly, the algorithm consists of two phases: the *LLM imitation learning* phase and the *standard RL* phase. In the first phase, the agent focuses on training its policy network, $\pi_\theta$, and value network, $Q_w$, to an effective state. Crucially, IHAC accelerates this process by leveraging high-level suggestions provided by a language model,

LLM, which significantly boosts learning efficiency, especially in the early stages when data samples are limited. We will provide further details on how LLM enhances the learning process later in the text. At the conclusion of the first phase, IHAC outputs a well-trained policy network, $\pi_\theta$, and value network, $Q_w$, both of which have been shaped effectively with the assistance of the LLM. In the second phase, IHAC begins using an existing RL algorithm, such as PPO, to continue refining the policy, starting from the already optimized $\pi_\theta$ and $Q_w$.

*Why two-phase?* Our two-phase algorithm design offers several advantages. First, it enables efficient token usage during training. Specifically, in our approach, the LLM is invoked only during the imitation learning phase, which constitutes a small portion of the overall learning process (in our experiments, for instance, it accounts for no more than 20 percent of the total epochs). This reduces the computational overhead from frequent LLM queries and minimizes overall token consumption. Second, the two-phase setup, along with key design components, allows for a dynamic trade-off between the LLM and the reinforcement learning (RL) approach. This structure combines the efficiency of the LLM with the superior effectiveness of the RL agent.

### 3.3 DETAILS OF IHAC

From here we discuss several key algorithm design innovations of our IHAC.

**High-Level Language Model Option** One of our key observations is that LLMs can provide effective high-level suggestions. For instance, if we provide the LLM with the current game status, such as "Agent sees a key and holds nothing," the LLM can generate general solutions based on its common-sense knowledge. However, utilizing these high-level solutions remains challenging, especially when the gap between high-level strategies and low-level actions is significant. To address this, we propose several algorithmic components, as shown in Line 5. We introduce a translator module, denoted as $\texttt{Trans}(\cdot) : \mathcal{S} \to \mathcal{L}$, which is an embedding model that maps any state $s$ to a language description $l$. For example, in the MiniGrid game, $\texttt{Trans}$ could take the abstract grid state as input and output a sentence describing the game status, such as "the player is in the corner with a knife in hand." Additionally, we use a predefined option module, denoted as $\texttt{ActionNet} : \mathcal{O} \times \mathcal{S} \to \mathcal{A}$, which infers an action $a$ based on both the high-level option and the current state. To summarize, as suggested in Line 5, we combine these elements into a composite LLM module, denoted as $\pi_{\texttt{LLM}}$, representing the LLM's guidance policy.

To improve efficiency, we limit the LLM's choices to a predefined set of options. This design reduces the complexity of the decision-making process, enabling more focused and relevant outputs. Additionally, the prompt structure ensures that the LLM's suggestions are reasonable and aligned with task goals, while achieving token efficiency by invoking the LLM only when necessary and restricting its output to predefined options.

**Annealing Strategy in Sampling** Given $\pi_{\texttt{LLM}}$, we now demonstrate how it can be leveraged to accelerate the RL agent's learning process during Phase I. We first focus on Line 6. Essentially, actions are selected by considering both the suggestions from the RL policy $\pi_\theta$ and the guidance from $\pi_{\texttt{LLM}}$. At stage $h$, the action $a_h$ is sampled as follows:

$$a_h \sim (1 - \lambda_t)\pi_\theta(\cdot|s_h) + \lambda_t \cdot \pi_{\texttt{LLM}}(\cdot|s_h).$$

Here, $0 < \lambda_t < 1$ is the imitation ratio, which represents the influence of $\pi_{\texttt{LLM}}$. Initially, $\lambda_t$ is set to a higher value, allowing the LLM's policy to significantly guide action selection. This aids in exploring the environment more effectively, especially when reward signals are sparse. As training progresses, $\lambda_t$ is gradually annealed, reducing the influence of $\pi_{\texttt{LLM}}$ and enabling the agent to rely increasingly on its own policy, $\pi_\theta$, which becomes more refined over time. This sampling design, combined with the annealing strategy, addresses challenges related to sparse rewards and ensures a smooth transition from exploration to exploitation. Ultimately, this approach enhances the reinforcement learning agent's performance.

**Accelerated Training Aided by LLM** Next we show that how to utilize the policy $\pi_{\texttt{LLM}}$ to assist the training process of the RL policy $\pi_\theta$ as well as its value $Q_w$, which greatly helps the later Phase II. Starting from Line 9, we have a buffer set $\mathcal{T}$ that stores all the past experiences so far.

Given the LLM-policy $\pi_{\text{LLM}}$, we update our RL agent $\pi_\theta$ and $Q_w$ by the standard imitation learning style. In detail, we update them by solving the following minimization problem:

$$\min_{\theta,w} \sum_{s,a,s',r\sim\mathcal{T}} (1-\alpha)\text{KL}(\pi_\theta(\cdot|s) \parallel \pi_{\text{LLM}}(\cdot|s))$$
$$+ \alpha \left[ Q_w(s,a) - (r + \gamma\bar{E}_{a'\sim\bar{\pi}_\theta(\cdot|s')}\bar{Q}_w(s',a')) - \text{KL}(\bar{\pi}_\theta(\cdot|s')\|\pi_{\text{LLM}}(\cdot|s')) \right]^2,$$

$$(1)$$

where $\text{KL}(\cdot\|\cdot)$ represents the KL-divergence, $0 < \alpha < 1$ represents the trade-off between the update of the policy network and the update of the value network, $\bar{\pi}_\theta, \bar{Q}_w$ suggest that these parameters are fixed during training. The KL regularizer term encourages $\pi_\theta$ to be closed to $\pi_{\text{LLM}}$, which is inspired by Zhang et al. (2024). Essentially speaking, we utilize LLM to guide the training for both our policy network and the value network. Although it seems that the value network $Q_w$ does not play its role during Phase I, we want to highlight that it serves as the value network in Phase II, which is standard for the actor-critic framework. Experimentally, we show that a high-quality value network serves an importanbt role in the final performance of the agent.

In general, our usage of $\pi_{\text{LLM}}$ as well as the structured training process, combined with careful token management, ensure that our reinforcement learning agent can efficiently learn from the LLM while minimizing resource usage.

## 4 EXPERIMENTS

In this section, we present the experimental setup and results to assess the effectiveness of our proposed method. We conducted experiments using the MiniGrid (Chevalier-Boisvert et al., 2023), NetHack (Küttler et al., 2020), and Crafter (Hafner, 2022) as our environments and we choose Prakash et al. (2024) and Zhou et al. (2024) as our baselines. See Appendix A for more details.

### 4.1 EXPERIMENT SETUP

**Baselines** To benchmark our approach, we compared IHAC against two related hierarchical reinforcement learning algorithms combined with LLM: **LLM4Teach** (Zhou et al., 2024) and **LLM×HRL** (Prakash et al., 2024). We made a few modifications to the original implementations to both baselines for a fair comparison, and we left the details to Appendix A.

**Component setup for IHAC** For `ActionNet`, these options are crafted based on domain knowledge and are intended to guide the LLM towards making decisions that are both effective and aligned with the task's objectives.There are two main types `ActionNet`, *Navigation* and *Interact*. We use A* algorithm to get the optimal action for agent going to the corresponding object. *Interact* contains all actions like "pick up" and "open". For `Trans` , we adapt the description method, which is consistent with previous methods (Prakash et al., 2024; Zhou et al., 2024). `Trans` takes the observation as the input. It will find all important items (e.g. key in MiniGird environment) and show the current condition of the agent (e.g. hp and weapon in NetHack environment). It can translate the observation to a prompt which will be given to the LLM to generate current option.

### 4.2 EXPERIMENT RESULTS FOR MINIGRID

**Environment description** We compare our IHAC with baselines on four distinct procedurally generated tasks within the MiniGrid environment (Chevalier-Boisvert et al., 2023): *SimpleDoorKey*, *TwoDoorKey*, *KeyInBox*, and *RandomDoorKey*. The agent is randomly placed in these environments, each with limited visibility. The objective is to explore the environment, find the correct key, and use it to unlock the door and exit. In *SimpleDoorKey*, the agent must find a single key and a door to unlock. *TwoDoorKey* presents multiple doors that the agent needs to unlock sequentially to find the exit. In *KeyInBox*, the key is hidden inside a box, requiring the agent to interact with the box to retrieve the key. *RandomDoorKey* adds an element of uncertainty, with the key randomly placed either inside a box or elsewhere in the environment.

**Environment adaptation** For MiniGrid, the LLM outputs a set of options: *go to target*, *pick up*, *drop*, *open*, *wait*, and *explore*. These options are selected based on the agent's current state. For example, if

the agent sees a box, it will choose to interact with it. Each option returns a distribution over specific actions, which is then used to guide the reinforcement learning agent. For all the approaches, we evaluate their policies every 20 iterations with 5 randomly generated testing seeds and report the averaged testing performance here. For some baselines, the models had not fully converged by the time they reached the predetermined number of iterations due to slower training speeds. However, to ensure a fair comparison with the other LLM-assisted models, we terminated training at the same iteration count for all models, even though some baselines had not yet converged, and included them in the experimental results. We use Vicuna 7b (Team, 2023) to conduct our experiments.

| Task | Method | Performance | | | |
| --- | --- | --- | --- | --- | --- |
| | | Avg. Step $l$ | Avg. Return $r$ | Avg. Success Rate $\beta$ | Consumed tokens |
| SDK | IHAC | $\mathbf{28.4 \pm 9.7}$ | $\mathbf{0.82 \pm 0.06}$ | $\mathbf{0.97 \pm 0.06}$ | $\mathbf{2.29 \times 10^8}$ |
| | LLM4Teach | $30.2 \pm 10.4$ | $0.81 \pm 0.07$ | $0.96 \pm 0.06$ | $7.13 \times 10^9$ |
| | LLM×HRL | $31.4 \pm 11.8$ | $0.79 \pm 0.08$ | $0.94 \pm 0.09$ | $3.32 \times 10^9$ |
| TDK | IHAC | $\mathbf{20.0 \pm 8.2}$ | $0.81 \pm 0.06$ | $\mathbf{0.96 \pm 0.06}$ | $\mathbf{1.92 \times 10^8}$ |
| | LLM4Teach | $33.9 \pm 10.4$ | $0.86 \pm 0.06$ | $0.95 \pm 0.07$ | $3.16 \times 10^9$ |
| | LLM×HRL | $21.1 \pm 9.5$ | $\mathbf{0.87 \pm 0.05}$ | $0.95 \pm 0.09$ | $2.11 \times 10^9$ |
| KIB | IHAC | $\mathbf{29.19 \pm 7.3}$ | $\mathbf{0.81 \pm 0.06}$ | $\mathbf{0.96 \pm 0.07}$ | $\mathbf{3.58 \times 10^8}$ |
| | LLM4Teach | $37.9 \pm 14.5$ | $0.77 \pm 0.09$ | $0.93 \pm 0.10$ | $5.97 \times 10^9$ |
| | LLM×HRL | $35.7 \pm 12.7$ | $0.80 \pm 0.08$ | $0.94 \pm 0.08$ | $4.95 \times 10^9$ |
| RBK | IHAC | $\mathbf{30.6 \pm 8.6}$ | $\mathbf{0.81 \pm 0.06}$ | $\mathbf{0.97 \pm 0.07}$ | $\mathbf{3.93 \times 10^8}$ |
| | LLM4Teach | $34.1 \pm 10.3$ | $0.79 \pm 0.06$ | $0.95 \pm 0.08$ | $7.36 \times 10^9$ |
| | LLM×HRL | $47.7 \pm 12.5$ | $0.77 \pm 0.07$ | $0.92 \pm 0.06$ | $5.81 \times 10^9$ |

Table 1: Performance of different methods on various tasks in MiniGrid environment. For average step term, smaller result means the better performance.

**Results** The main results are shown in Figure 9. Our experimental results primarily compare the average steps needed for success, the reward rate, and the success rate. Our model performed better in all four environments with sparse rewards, which means it can get high and stable return rate and success rate with fewest number of iterations. At the same time, the steps it used also decrease quickly and it can even surpass the benchmark set by other baseline models. Our model demonstrates a significant advantage compared to existing large model-assisted reinforcement learning models. Especially in more complex environment, the advantage of our model will be more obvious. We analyze the results in detail.

- **KeyInBox(KIB)** and **TwoDoorKey(TDK)**: In these two environment, our model achieved good performance after 2,500 iterations, while the other models converged to optimal performance around 4,500 iterations, which is 1 times more than our method.
- **SimpleDoorKey(SDK)**: We can observe that in this environment, the curves of the three models are very close to each other. IHAC shows a slight acceleration effect. This is because the environment is quite simple, and the agent only needs to directly obtain the key and open the door, resulting in good performance from all three models.
- **RandomBoxKey(RBK)**: This task requires the agent to open the box, retrieve the key, and finally open the door. However, the presence of the key in the box is random, which adds to the difficulty of the task. Our model performs exceptionally well, showing a significantly faster convergence speed compared to other models, and it is able to complete the task with fewer steps.

### 4.3 EXPERIMENT RESULTS FOR NETHACK

**Environment description** We compare performance on two procedurally generated tasks in the NetHack environment (Samvelyan et al., 2021): *LavaCross* and *Monster*. In *LavaCross*, the agent must drink a potion of flight from its inventory and fly over a lava pit to reach the exit. The challenge lies in correctly identifying and using the potion at the right moment while avoiding the lava. In *Monster*, the agent explores a 10x10 room while being chased by two monsters.

**Environment adaptation** We use the embedded NetHack environment provided by Goodger et al. (2023). The NLE (NetHack Learning Environment) translates symbolic states into natural language, functioning as the `Trans` module in this specific environment. Additionally, it provides relevant options based on the current state. Thus, the LLM only needs to choose a rational option from the list of provided options. Each of these options corresponds to a distribution over specific actions, from which the reinforcement learning agent samples to execute in the environment. For our experiments, we utilized ChatGPT-3.5-turbo (Ye et al., 2023). We set the scenario by providing the model with a detailed description of the task it was about to undertake. We evaluate our model every 10 iterations with 5 randomly generated testing seed.

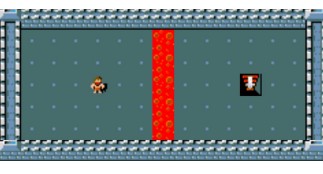
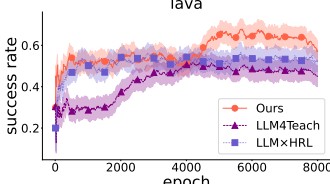
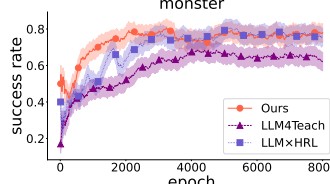

Figure 2: Nethack environment screenshot

Figure 3: Lavacross environment result

Figure 4: Monster environment result

**Results** The experimental results are shown in Figures 10 and 11. Overall, IHAC outperforms the two baselines. A detailed analysis is provided below:

- **LavaCross**: IHAC trains slightly faster than the other two algorithms. The final success rate is similar, but when comparing token usage, we observed that since IHAC stops using the language model after the imitation phase, the number of tokens spent is significantly reduced.

- **Monster**: In this experiment, during the early stages of training, IHAC's training speed is faster than the other two algorithms, achieving a higher success rate in a shorter amount of time. When the model converges, IHAC's success rate is higher than the other two algorithms, with IHAC outperforming LLM4Teach significantly and also surpassing LLM×HRL.

In the lava environment, IHAC achieves a higher success rate compared to both LLM×HRL and LLM4Teach, with success rates of IHAC improves the success rate by **14.75%** over LLM×HRL and by **21.31%** over LLM4Teach. Additionally, IHAC consumes significantly fewer tokens, which is **90%** less than LLM×HRL and **95%** less than LLM4Teach.

### 4.4 EXPERIMENT RESULTS FOR CRAFTER

**Environment description** Crafter (Hafner, 2022) is a 2D version of the Minecraft environment, featuring the same complex state space but with a smaller observation space. Similar to Minecraft, it requires the agent to gather resources and craft tools to survive in a hazardous environment. We focus on one hierarchical task in Crafter: *MakeStonePickaxe*. In this task, the agent must first collect enough wood, craft a workbench, and then use the workbench to create a wooden pickaxe. After acquiring the wooden pickaxe, the agent uses it to gather stone and finally crafts a stone pickaxe at the workbench. Therefore, there are five achievements involved in each game that the agent aims to complete. See Figure 5 for an example.

**Environment adaptation** We apply (Moon et al., 2023) as our basic network structure as Hafner (2022) shows that basic PPO algorithm performs not satisfactorily in relatively easy achievements and hardly accomplishes difficult achievements implemented in Crafter environment. Check Table A.1 in Hafner (2022) for further detailed results. In our experiment, each iteration will have 8 parallel experiments running simultaneously, and every 5 iterations will be evaluated once.

**Results** From Figure 6, we observe that our proposed IHAC method consistently outperforms both baseline algorithms in terms of success rate across all tasks:

Figure 5: Crafter example
ple

| Task | IHAC | LLMxHRL | LLM4Teach |
|---|---|---|---|
| Collect Stone | **67.14 ± 7.98** | 55.68 ± 8.04 | 10.42 ± 2.56 |
| Make Wood Pickaxe | **82.60 ± 3.02** | 73.76 ± 3.56 | 53.42 ± 4.98 |
| Make Stone Pickaxe | **13.64 ± 1.47** | 4.80 ± 2.64 | 2.08 ± 0.91 |
| Collect Wood | 95.65 ± 0.59 | **96.19 ± 1.34** | 74.03 ± 2.64 |
| Place Table | **95.24 ± 0.25** | 86.15 ± 1.67 | 63.82 ± 2.89 |
| Consumed Tokens | $\mathbf{6.83 \times 10^5}$ | $3.57 \times 10^7$ | $5.26 \times 10^7$ |

Figure 6: Success rates and consumed tokens in Crafter.

- **Collect wood**: All methods achieve a near 100% success rate, indicating that this is a relatively straightforward task for hierarchical reinforcement learning agents.

- **Collect stone**: While all methods continue to perform well, there is a noticeable slight drop in the success rate for the LLM4Teach approach. Our method maintains a high success rate, showing robustness even in more complex tasks.

- **Make wood pickaxe** and **Make stone pickaxe**: As the tasks become more complex, we see that IHAC remains the most reliable, with LLM×HRL showing competitive performance. However, LLM4Teach struggles to maintain a comparable success rate, particularly in crafting the stone pickaxe, likely due to its inefficient exploration during earlier stages.

- **Place table**: This task requires multiple sequential steps, and IHAC continues to show superior performance, demonstrating its ability to handle more complex hierarchical tasks.

One of the key advantages of the IHAC algorithm is its efficient use of tokens. As shown in the final bar of the graph (Figure 6), IHAC consumes significantly fewer tokens compared to both LLM4Teach and LLM×HRL. This is a critical factor, as token consumption directly correlates with computational cost and efficiency. Our IHAC algorithm employs LLMs only during the early imitation learning phase, where LLM guidance is used once per step. As detailed results shown in Appendix 6, in difficult and long-term environment, IHAC only consumes less than **2%** of the tokens consumed by LLM×HRL, and nearly **1%** of the tokens used by LLM4Teach. This is because IHAC only queries the LLM during the IL phase, which spans the first fifth of the total training iterations.

In contrast, LLM4Teach lacks high-level sampling guidance, resulting in extended training steps. Additionally, it queries the LLM five times per step to generate the option distribution, leading to significant token consumption. LLM×HRL, while using a sampling policy to prevent the agent from acting inefficiently, also faces high token usage due to the complex environment. Specifically, 11 different options (detailed in Appendix B.4) are presented for the LLM to choose from at each step, requiring the agent to query the LLM 11 times per step to obtain the option distribution. This frequent querying is the primary factor driving up token consumption. Both LLM×HRL and LLM4Teach query the LLM throughout the entire training process, resulting in a large token overhead. Detailed baseline prompt designing methods are shown here Appendix A. Our approach, on the other hand, enables the model to rapidly learn an initial policy, effectively addressing the sparse reward problem and dramatically reducing token usage in later stages.

## 4.5 ABLATION STUDY

In this section, we conduct the ablation study of IHAC. We compare IHAC with several of its variants to suggest the effectiveness of each component IHAC has. We select the MiniGrid environment with the KeyInBox task to compare. The results are in Figure 7. The methods compared include the base model, an optimized prompt model, and several variations with different loss functions and sampling strategies. See 8 for more details. Below is a summary of the results for each model:

- **(I) Base Model (LLM×HRL):** This model employs a fixed sampling policy with standard PPO updates during training, without the application of imitation learning. In the early stages, we observe that the Base Model requires more time steps to learn how to complete the task.

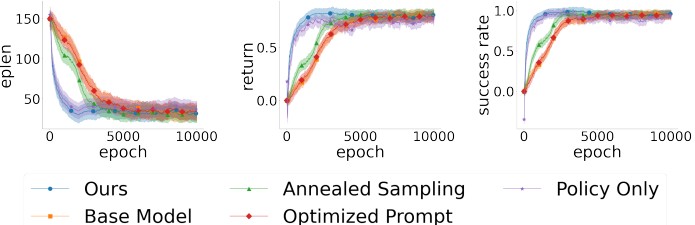

Figure 7: Success Rate, Return & Eplen of Minigrid KeyInBox environment

|     | (I) | (II) | (III) | (IV) | (V) |
|-----|-----|------|-------|------|-----|
| NP  | ✗   | ✓    | ✓     | ✓    | ✓   |
| NS  | ✗   | ✗    | ✓     | ✓    | ✓   |
| IL  | ✗   | ✗    | ✗     | ✓    | ✓   |
| TV  | ✗   | ✗    | ✗     | ✗    | ✓   |

Figure 8: Ablation studies with different components enabled.

- **(II) NP:** This variant differs from the Base Model by incorporating a new prompting method, similar to IHAC, while still maintaining a fixed sampling policy and traditional PPO updates, without imitation learning. In this experiment, the Optimized Prompt performs similarly to the Base Model, with a slightly higher success rate. This is because the optimized prompt does not significantly alter the outcome of the LLM.

- **(III) NP+NS:** This model adopts the IHAC sampling policy throughout the entire training process but does not include an imitation learning phase. In this experiment, Annealed Sampling outperforms the Optimized Prompt, achieving a higher success rate and faster training speed. This is because the decaying influence on sampling allows the model to sample according to the real distribution, which helps maintain the convergence of the algorithm.

- **(IV) NP+NS+IL:** This experiment includes an imitation learning phase but differs from our approach by applying only the policy loss. By the RL phase, the policy network is trained, but the value network remains unchanged. The annealed sampling strategy is used during imitation learning, and actions are sampled directly from the policy network during the RL phase. In this experiment, we observe that without training the $V$ network, more steps are required to train it during the PPO phase. Its performance is very similar to, but slightly worse than, our algorithm.

- **(V) NP+NS+IL+TV (Proposed Model):** Our model combines both value and policy losses, along with annealed sampling during the imitation learning phase. The IL phase enables the network to quickly learn a baseline policy. With additional imitation learning on the $V$ network, the transition from IL to RL is smoother. As a result, our model achieves the best performance among all tested algorithms.

## 5 CONCLUSION

In this work, we addressed the challenges of solving complex tasks with sparse rewards by proposing a novel two-phase training framework that combines imitation learning and reinforcement learning. Our approach efficiently leverages LLMs during the early imitation learning phase, allowing the agent to rapidly acquire foundational skills and significantly accelerate reinforcement learning. A key contribution of our framework is the strategic use of LLMs restricted to the pre-training phase, which substantially reduces token consumption while maintaining strong performance. This lightweight and efficient design balances the powerful reasoning capabilities of LLMs with practical resource constraints, making it suitable for real-world applications. Additionally, the integration of a hierarchical structure combining value-based and policy-based learning enables faster convergence and better task generalization. Techniques such as annealed sampling and adaptive policy training further enhance learning efficiency by ensuring a smooth transition from LLM-guided exploration to agent-driven exploitation. Our experimental results demonstrate that the proposed framework achieves superior performance with improved sample efficiency and reduced computational costs compared to baseline methods. Future work will focus on extending this framework to more complex environments and exploring advanced strategies to further reduce reliance on LLMs.

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

## A  BASELINE MODIFICATION

We modify these baselines. In LLM4Teach, the prompt consisted solely of the observation (obs), leading to highly open-ended responses from the LLM, which made it challenging to extract useful information. To enhance stability, we modified the prompt by appending "Choose an option from [$o_i$, ... ]" at the end, which is the same as our format. LLM×HRL originally relied on a pre-trained neural network for skills, which required substantial engineering effort and did not guarantee reliable outputs. We replaced the neural network with pre-defined skill functions to maintain consistency with our own algorithm. In general, all these algorithms use the same logic from `Trans` and `ActionNet`, ensuring the fairness. To be more specific, given fixed current state, all these three methods will call the same `Trans` function to get the corresponding option and `ActionNet` will output the higher-level action distribution for the next operation. The only difference between IHAC and baselines is how to utilize the higher-level action distribution.

## B  DETAILED EXPERIMENT SETTINGS

### B.1  COMPUTATIONAL RESOURCES

All our experiments are conducted on two NVIDIA A6000 GPUs for training, and we utilize PyTorch as our primary deep learning framework Paszke et al. (2019). In MiniGrid, we use Vicuna 7b Team (2023) as the higher policy model. In NetHack and Crafter, it is too difficult for Vicuna to infer the current state, so we apply ChatGPT 3.5-turbo Ye et al. (2023) instead.

### B.2  MINIGRID SETTINGS

MiniGrid is a terrific environment Chevalier-Boisvert et al. (2023) for applying HRL because its natural tasks can easily be divided into sub-tasks. In our work, we implement $SimpleDoorKey$, $TwoDoorKey$, $KeyInBox$, and $RandomBoxKey$. We use two NVIDIA A6000 GPUs for training.

**Hyperparameters**

We use PPO as the base algorithm for the controller Zhou et al. (2024). The actor and critic networks in MiniGrid share a simple and effective architecture. The policy network (actor) is a two-layer fully connected neural network that maps the input embedding to the action space. The first layer consists of 64 hidden units with ReLU activation, followed by an output layer with the dimensionality equal to the size of the action space. Similarly, the value network (critic) shares a similar structure, with the final output being a single scalar representing the state value. We list all the parameters involved in the RL and IL training below.

| Variable | Value |
|---|---|
| Number of trajectories per iteration | 10 |
| Number of epochs per iteration | 3 |
| Minibatch size | 128 |
| Entropy loss coefficient | 0.001 |
| Value function loss coefficient | 0.5 |
| Discount factor | 0.99 |
| Learning rate | 0.001 |
| Clipping parameter | 0.2 |
| Maximum gradient norm | 0.5 |

Table 2: RL hyperparameters in MiniGrid experiments.

**Prompts**

Below are four examples of prompt design for MiniGird, corresponding to four different tasks respectively.

| Variable | Value |
|---|---|
| Number of trajectories per iteration | 10 |
| Number of epochs per iteration | 3 |
| Minibatch size | 128 |
| Value function loss coefficient ($\alpha$) | 0.25 |
| Entropy loss coefficient (1-$\alpha$) | 0.75 |
| Learning rate | 0.001 |
| KL divergence coefficient in Q | 0.5 |
| Maximum gradient norm for q | 0.5 |
| Maximum value for q | 2 |
| Sampling Discount factor | 0.99 |
| Sampling Update Interval | 10 |
| Pretraining Percentage | 10% |

Table 3: IL pretraining hyperparameters in MiniGrid experiments.

---

**SimpleDoorKey Example Prompt**

**Problem title** : SimpleDoorKey
**Description** : In a locked 2D grid room, there is an agent whose task is to open the door. The door can only be opened while agent holds the key. The agent can perform the following actions: explore, go to its goal, pick up its goal, drop its carrying object, or open its goal. You need to minimize the step to open the door. Your response should include your reason and follow the format "Answer : (Your Choice)".
**Example**:
Observation : Agent see a key and holding nothing. Choose an option from ["explore", "go to key, pick up key"].
Answer : Explore.

---

**TwoDoor Example Prompt**

**Problem title** : TwoDoor
**Description** : In a locked 2D grid room, there is an agent whose task is to open the door. The door can only be opened while agent holds the key. There are two different doors and you only need to open one of them. The agent can perform the following actions: explore, go to its goal, pick up its goal, drop its carrying object, open its goal. You need to minimize the step to open the door. Your response should include your reason and follow the format "Answer : (Your Choice)".
**Example**:
Observation : Agent see door 1, door 2, key, hold nothing. Choose an option from ["explore", "go to key, pick up key", "go to door 1, open door 1", "go to door 2, open door2"]
Answer : Go to key, pick up key.

---

**KeyInBox Example Prompt**

**Problem title** : KeyInBox
**Description** : In a locked 2D grid room, there is an agent whose task is to open the door. The door can only be opened while agent holds the key. And the key is in a box. The agent can perform the following actions: explore, go to its goal, pick up its goal, drop its carrying object, open its goal, or toggle its goal. You need to minimize the step to open the door. Your response should include your reason and follow the format "Answer : (Your Choice)".
**Example**: Observation : Agent see a box and holding nothing. Choose an option from ["explore", "go to box, toggle the box"]
Answer : go to the box, toggle the box.

---

**RandomBoxKey Example Prompt**

**Problem title** : RandomBoxKey
**Description** : In a locked 2D grid room, there is an agent whose task is to open the door. The door can only be opened while agent holds the key. The key could be outside the box or inside the box. The agent can perform the following actions: explore, go to its goal, pick up its goal, drop its carrying object, open its goal, or toggle its goal. You need to minimize the step to open the door. Your response should include your reason and follow the format "Answer : (Your Choice)".
**Example**:
Observation: Agent sees a box and a key, holding nothing. Choose an option from ["explore", "go to box, toggle the box", "go to key, pick up key"].
Answer: Go to key, pick up key.

---

**`Trans` and `ActionNet`**

We maintain the same `Trans` and `ActionNet` setting as Zhou et al. (2024). We primarily use five different options: explore, go to, pick up, drop, and open.

- $explore$: When the agent chooses explore as the current option, it will explore the currently unseen tiles.
- $goto$: This option takes an item as input (e.g., "go to door") and generates actions using the A* algorithm until the agent reaches the goal.
- $pickup$: This is a one-step action to pick up the front item.
- $drop$: This is a one-step action to drop the item that the agent is holding.
- $open$: This is a one-step action to open the front item, for example the door and the box.

**Experiment Result in Detail**

We provide the detailed asymptotic performances for all tasks in Table 3. The Minigrid results are averaged over 5 tests runs.

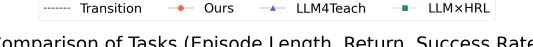

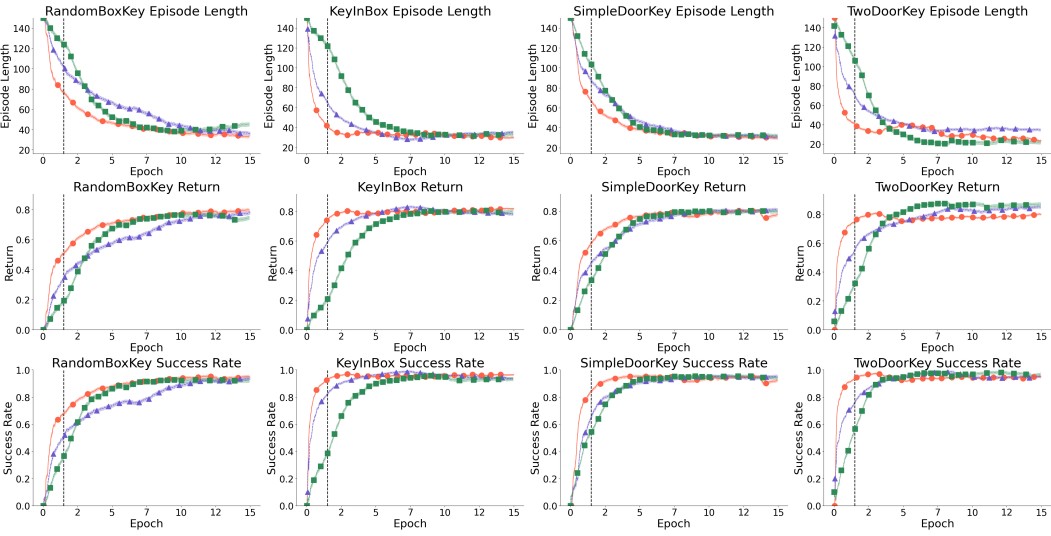

Figure 9: The tested average returns, task completion success rates, and steps (eplen) vs. the training iteration index of the compared methods across four environments. Our model uses just 1/10 of the tokens required by the other two models. We use dashed lines to clearly mark the transition between the imitation learning phase and the reinforcement learning phase. From the graphs, it is clear that in the later stage of reinforcement learning, the model's performance continues to improve.

### B.3 NETHACK SETTINGS

**Hyperparameters**

We use PPO as the base algorithm for the controller Schulman et al. (2017), and we list all the parameters involved in the RL and IL below. We follow Küttler et al. (2020) to set the backbone architecture, which adapts CNN LeCun et al. (1998) for both the actor and critic networks.

| Variable | Value |
|---|---|
| Number of trajectories per iteration | 10 |
| Number of epochs per iteration | 3 |
| Minibatch size | 128 |
| Entropy loss coefficient | 0.001 |
| Value function loss coefficient | 0.5 |
| Discount factor | 0.99 |
| Learning rate | 0.001 |
| Clipping parameter | 0.2 |
| Maximum gradient norm | 0.5 |

Table 4: RL hyperparameters in Nethack experiments.

| Variable | Value |
|---|---|
| Number of trajectories per iteration | 10 |
| Number of epochs per iteration | 3 |
| Minibatch size | 128 |
| Value function loss coefficient ($\alpha$) | 0.25 |
| Entropy loss coefficient (1-$\alpha$) | 0.75 |
| Learning rate | 0.001 |
| KL divergence coefficient in Q | 0.5 |
| Maximum gradient norm for q | 0.5 |
| Maximum value for q | 2 |
| Sampling Discount factor | 0.97 |
| Sampling Update Interval | 10 |
| Pretraining Percentage | 10% |

Table 5: IL pretraining hyperparameters in Nethack experiments.

**Prompts**

Below are two prompt examples for *LavaCross* and *Monster*.

**LavaCross Example Prompt**

**Problem title** : LavaCross
**Description** : You are a game agent in the Nethack environment. Your goal is to drink the potion and cross lava. First, you need to drink the potion which is already in your inventory. Next, you need to cross lava lake and enter the exit. You need to minimize the step to open the door. Your response should include your reason and follow the format "Answer : (Your Choice)".
**Example**:
Observation: "You have a +1 club (weapon in hand)", "You have a +2 sling (alternate weapon; not wielded)", "You have 19 uncursed flint stones (in quiver pouch)", "You have 29 uncursed rocks", "You have an uncursed +0 leather armor (being worn)", "Strength: 18/18", "Dexterity: 15", "Constitution: 16", "Intelligence: 7", "Wisdom: 9", "Charisma: 10", "Depth: 1", "Gold: 0", "HP: 16/16", "Energy: 2/2", "AC: 8", "XP: 1/0", "Time: 7", "Position: 39|8", "Hunger: Not Hungry", "Monster Level: 0", "Encumbrance: Unencumbered", "Dungeon Number: 0", "Level Number: 1", "Score: 10", "Alignment: Neutral", "Condition: Levitating", "You see a vertical wall far west", "You see a vertical wall near east", "You see a southeast corner near southeast", "You see a horizontal wall near south and southwest", "You see a area of lava near south southwest", "You see a stairs up near west southwest", "You see a stairs down very near east", "You see a lava very near south southwest", "You see a horizontal wall adjacent north, northeast, and northwest", "You see a lava adjacent southwest and west".
Choice : ["move north", "move south", "move east", "move west", "move northwest", "move northeast", "move southwest", "move southeast"]
**Answer**: "move southwest"

**Monster Example Prompt**

**Problem title** : Monster
**Description** : You are a game agent in the Nethack environment. Your goal is to leave the room by killing monsters. You need to minimize the step to open the door. Your response should include your reason and follow the format "Answer : (Your Choice)".
**Example**:
Observation: "You have a +0 short sword (weapon in hand)", "You have 14 +0 daggers (alternate weapon; not wielded)", "You have an uncursed +1 leather armor (being worn)", "You have an uncursed potion of sickness", "You have an uncursed lock pick", "You have an empty uncursed sack", "Strength: 15/15", "Dexterity: 15", "Constitution: 10", "Intelligence: 11", "Wisdom: 15", "Charisma: 9", "Depth: 1", "Gold: 0", "HP: 12/12", "Energy: 2/2", "AC: 7", "XP: 1/0", "Time: 2", "Position: 37|9", "Hunger: Not Hungry", "Monster Level: 0", "Encumbrance: Unencumbered", "Dungeon Number: 0", "Level Number: 1", "Score: 0", "Alignment: Chaotic", "Condition: None", "You see a stairs down near east", "You see a dark area near east, southeast, and south", "You see a dark area very near southwest and west", "You see a dark area adjacent north, northeast, and northwest", "You see a kobold adjacent south",
Choice : ["move north", "move south", "move east", "move west", "move northwest", "move northeast", "move southwest", "move southeast, "attack the kobold"]
**Answer**: "attack the kobold."

**Trans and ActionNet**

We use the same setting fromTupper (2023). It can translate the observation from NetHack environment to a natural language as the input prompt. it will evaluate each action and choose the best one as the high level action. We list options involved in Nethack as below:

- *goto*: This option takes an item as input (e.g., "go to weapon") and generates actions using the A* algorithm until the agent reaches the goal.
- *interact*: This is a one-step action including all interactions with items, for example attack, drink and etc.

**Detailed Results**

The table below shows the success rate for different achievements in the process of lavacross and monster environment and the total tokens consumed during training.

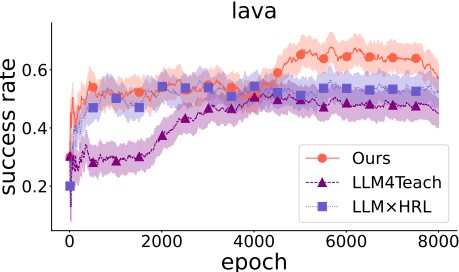
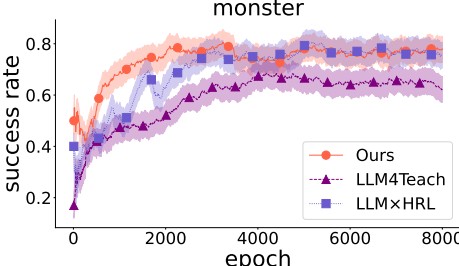

Figure 10: Lavacross environment result
Figure 11: Monster environment result

### B.4 CRAFTER SETTINGS

Crafter Hafner (2022) contains 22 different achievements, including "wake up" and "eat cow." We focus on making a stone pickaxe. To accomplish this task, the agent must collect enough wood, make a table, use the table to make a wood pickaxe, use the wood pickaxe to collect stones, and then make the stone pickaxe on the table.

**Hyperparameters**

In this controller model, we use ResNet He et al. (2016) as our backbone structure , following the settings of Moon et al. (2023). The tables below show our main hyperparameters for the experiments.

| Variable | Value |
|---|---|
| Number of epochs per iteration | 3 |
| Minibatch size | 8 |
| Clipping parameter | 0.2 |
| Value function loss coefficient | 0.5 |
| Entropy loss coefficient | 0.01 |
| Learning rate | 0.0003 |
| Maximum gradient norm | 0.5 |
| Auxiliary update frequency | 8 |
| Number of auxiliary epochs | 6 |
| Policy distribution coefficient | 1.0 |
| Value function distribution coefficient | 1.0 |
| Auxiliary KL loss coefficient | 1.0 |
| Auxiliary sampling weight decay | 0.95 |

Table 6: RL hyperparameters in Crafter experiments.

| Variable | Value |
|---|---|
| Number of steps per iteration | 512 |
| Number of processing units | 8 |
| Learning rate | 0.0003 |
| Value function loss coefficient ($\alpha$) | 0.25 |
| Entropy loss coefficient (1-$\alpha$) | 0.75 |
| Sampling Discount factor | 0.95 |
| Sampling Update Interval | 10 |
| Sampling weight initial value | 1.0 |
| KL divergence coefficient in Q | 0.5 |
| Maximum Step length | 512 |
| Pretraining Percentage | 20% |

Table 7: IL pretraining hyper parameters in Crafter experiments.

**Prompts**

There are two prompt temples for Crafter tasks.

---

Crafter Example Prompt

**Problem title** : MakeWoodPickaxe
**Description** :  You are a game agent in the Crafter environment. Your goal is to make a wood pickaxe. First, you need to collect four woods. Next, you need to build a table to make a wood pickaxe. Then, you should use the wood to make the stone pickaxe.
**Observation**:
Observation: Agent sees grass, coal, tree, stone, path, sand, table, water. You have 1 wood, 2 stone, 2 wood_pickaxe. Choose an option from ["attack zombie", "attack skeleton", "drink water", "eat cow", "sleep", "chop tree", "get stone", "craft wood_pickaxe", "craft stone_pickaxe", "build table", "explore"].
**Answer** : craft wood_pickaxe.

---

Crafter Example Prompt

**Problem title** : MakeStonePickaxe
**Description** :  You are a game agent in the Crafter environment. Your goal is to make a stone pickaxe. First, you need to collect four woods. Next, you need to build a table to make a wood pickaxe. Then, you should use the wood pickaxe to get a stone. Finally, you should get back to the table and make the stone pickaxe.
**Example**:
Observation: Agent sees grass, coal, tree, stone, path, sand, water. You have 1 wood. You are thirsty now. Choose an option from ["attack zombie", "attack skeleton", "drink water", "eat cow", "sleep", "chop tree", "get stone", "craft wood_pickaxe", "craft stone_pickaxe", "build table", "explore"].
**Answer** : drink water.

---

**Trans and ActionNet**

We design `Trans` and `ActionNet` similar to MiniGrid B.2. Here are basic options used in Crafter:

- *explore*: When the agent chooses explore as the current option, it will explore the currently map.
- *goto*: This option takes an item as input (e.g., "go to wood") and generates actions using the A* algorithm until the agent reaches the goal.
- *collect*: This is a one-step action to collect the front item.
- *drop*: This is a one-step action to drop the item that the agent is holding.
- *build*: This is a one-step action to build the table, wood pickaxe or stone pickaxe.

- $attack$: This is a one-step action to attack the front enemy.
- $drink$: This is a one-step action to drink water.
- $sleep$: This is a one-step action to sleep.

## C  SENSITIVITY ANALYSIS

In this section, we will do some sensitivity analysis in the MiniGrid Simple Door Key environment to additional parameters used in our two-phase training process, including Pretraining Percentage ($p$), the ratio ($r = \alpha/(1 - \alpha)$) between value function loss coefficient and entropy loss coefficient, and the updating method of $\lambda_t$.

**Pretraining Percentage ($p$)** $p$ represents the proportion of total iterations dedicated to the imitation learning phase. The base setting for $p$ is 10%, and we tested alternative settings of 5% and 15%. The results, as shown in Figure 12, include comparisons across three metrics: Episode Length, Return, and Success Rate. From the results, we observe that the choice of $p$ has minimal impact on performance, as it nearly does not affect the performance or the convergence of IHAC. This analysis suggests that

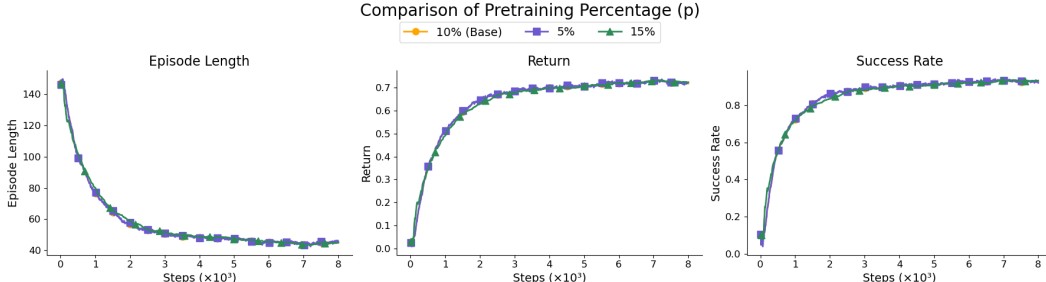

Figure 12: Comparison of Pretraining Percentage ($p$) in MiniGrid Door Key environment.

our method is robust to changes in $p$, providing flexibility in adjusting the duration of the imitation learning phase without significantly compromising performance.

**Ratio ($r$)** To analyze the effect of varying the ratio ($r$) between the value function loss coefficient and the entropy loss coefficient, we conducted experiments with three different configurations: $r = 1 : 3$ (base configuration), $r = 1 : 1$, and $r = 3 : 1$. The results, presented in Figure 13, evaluate performance across three metrics: Episode Length, Return, and Success Rate.

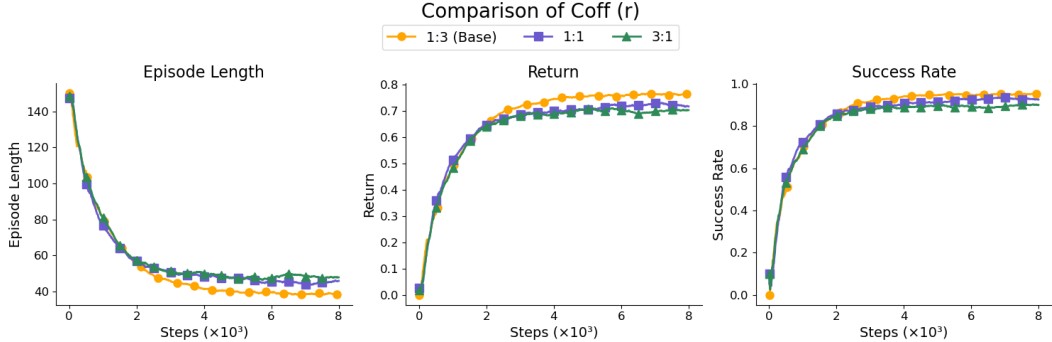

Figure 13: Comparison of different ratios ($r$) between value function loss and entropy loss coefficients on MiniGrid Simple Door Key environment.

The results suggests increasing the weight of the policy loss ($r = 1 : 3$) results in improved performance, with shorter episode lengths, higher returns, and greater success rates compared to other configurations. These findings indicate that prioritizing the policy loss over the entropy loss allows the model to optimize decisions more effectively, leading to better sample efficiency and task completion metrics.

**Updating Method of** $\lambda_t$ To evaluate the sensitivity of our method to the updating strategy for $\lambda_t$, we conducted experiments with $\lambda_t \in \{0.99^t, 0.95^t, 0.75^t\}$. Here $\lambda_t$ denotes the decay factor applied to anneal sampling. The results, shown in Figure 14, evaluate the performance across Episode Length, Return, and Success Rate.

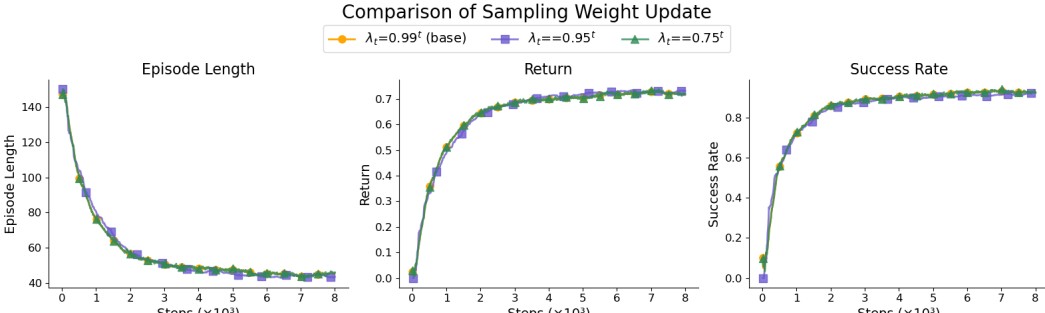

Figure 14: Comparison of different updating methods for $\lambda_t$ in the MiniGrid Simple Door Key environment.

From the results, we observe that IHAC is robust to the choice of $\lambda_t \in \{0.99^t, 0.95^t, 0.75^t\}$ as they share nearly the same performance. Combining with the ablation study of $\lambda_t$ in Section 4.5, we claim that IHAC works as long as a large $\lambda_t$ is selected.

Generally speaking, while our method introduces additional parameters, the tuning process remains straightforward and these parameters have minimal impact on the overall experimental results. This demonstrates the robustness of our approach, as the model performs well across a range of parameter configurations. Notably, we observed that increasing the weight of the policy loss significantly improves model training, which aligns with our intuition. By prioritizing the training of the policy, we can achieve better performance, reinforcing the importance of focusing on optimizing the policy for more effective decision-making.

