# OpenReview forum: "Leveraging Imitation Learning and LLMs for Efficient Hierarchical Reinforcement Learning"
_ICLR.cc/2025/Conference — Submitted to ICLR 2025_

### Official Review · Reviewer_LwQs · 2024-10-20

**Soundness:** 3
**Presentation:** 3
**Contribution:** 2
**Rating:** 6
**Confidence:** 3

**Summary:**

The IHAC framework models decision-making as a hierarchical RL problem, utilizing a two-phase approach:
- In the first phase, it uses an external LLM for imitation learning, guiding the selection of high-level options to accelerate early learning when the agent's experience is limited  (in both exploration and exploitation side)
- In the second phase, a standard RL algorithm like PPO further refines the policy

Tested on benchmarks like MiniGrid, IHAC outperforms existing methods in efficiency and performance, especially in optimizing LLM token usage.

**Strengths:**

- The paper is well-motivated, since the LLM policy can be potentially informative, it can be used to faciliate both exploration and exploitation procedure via policy mixing and KL regularization
- Comprehensive experiment is conducted, provide a clear picture of practical performance

**Weaknesses:**

- My main concern about this paper is it's novelty:
1. To ground LLM as a policy planner in real environment, modelling the decison problem as hierarchical RL is well-discussed in literature (e.g., https://arxiv.org/pdf/2310.10645)
2. Though this work also uses LLM to help learning a RL policy beyond simple imitation, but the key regularized based method, as pointed in the paper, is discussed in https://arxiv.org/pdf/2402.16181

Hence, it seems that the main contribution is to leverage LLM in the exploration stage. I would suggest authors to better discuss and highlight the contributions

**Questions:**

Despite the weakness, I also have the questions:
1. Can authors provide a more detailed discussion on the importance and effectiveness of using LLM policy in exploration and exploitation respectively? How such design sampling / regularization contributes to the better sample efficiency?

---

> ### Author Response · Authors · 2024-11-25
>
> Thank you for your detailed comments.
>
> **Q1**: Novelty. To ground LLM as a policy planner in real environment, modeling the decision problem as hierarchical RL is well-discussed in literature. Though this work also uses LLM to help learning a RL policy beyond simple imitation, but the key regularized based method has been discussed in previous work.
>
> **A1**:
> We acknowledge that using hierarchical RL to model decision problems for grounding LLMs as policy planners has been discussed in prior literature, such as the work referenced. In response, we have updated the Related Works section of our paper to incorporate a more comprehensive discussion of these studies like [1]. While previous studies like [2] provide valuable insights about our adapted regularization methods, our approach focuses on a lightweight, option-based framework that utilizes predefined high-level options rather than dynamically generating high-level actions. Our method also differs by integrating imitation learning and reinforcement learning in a two-phase process to address computational efficiency and token usage, which we believe is a novel contribution to this area.
> [1] Li, B., Wu, P., Abbeel, P., & Malik, J. (2023). Interactive task planning with language models. arXiv preprint arXiv:2310.10645.
> [2] Zhang, S., Zheng, S., Ke, S., Liu, Z., Jin, W., Yuan, J., ... & Wang, Z. (2024). How Can LLM Guide RL? A Value-Based Approach. arXiv preprint arXiv:2402.16181.
>
>
>
>
>
> **Q2**: it seems that the main contribution is to leverage LLM in the exploration stage. I would suggest authors to better discuss and highlight the contributions
>
> **A2**: To clarify, the main contribution of our work extends beyond simply leveraging light-weight LLMs in the exploration stage. In detail, IHAC incorporates several novel design elements. First, it uses separate learning phases to effectively leverage LLM guidance during the initial stage and refine the policy through reinforcement learning in the later stage. Second, it employs an adaptive sampling strategy that dynamically combines inputs from both the LLM and the RL agent during imitation learning. Third, it introduces KL-regularized terms applied to both the policy and value functions, which enhance stability and efficiency during training. We hope this clarification addresses your concerns and highlights the broader scope of our contributions.
>
>
> **Q3**: Can authors provide a more detailed discussion on the importance and effectiveness of using LLM policy in exploration and exploitation respectively? How such design sampling / regularization contributes to the better sample efficiency?
>
>
> **A3**:
> We would like to clarify the distinct roles of the LLM policy in exploration and exploitation within our framework. For exploration, the LLM policy plays a critical role in accelerating exploration, especially during the imitation learning phase. By leveraging the LLM’s general knowledge and reasoning abilities, the agent is guided toward high-level actions that are more likely to yield rewards or progress in the task. This significantly reduces the time spent exploring irrelevant or suboptimal trajectories, which is particularly beneficial in sparse-reward or complex environments where standard exploration techniques often struggle.
>
> For exploitation, we assume that you are asking how does IHAC leverage the information collected so far. In fact, the agent relies entirely on its own learned policy to select actions. This transition from LLM-guided exploration to independent exploitation ensures that the policy becomes fully autonomous and is capable of optimizing performance without continued reliance on the LLM. This design not only minimizes computational and token costs but also ensures that the final policy adapts effectively to the task environment.

---

> ### Author Response · Authors · 2024-11-25
>
> **Q4**: How such design sampling / regularization contributes to the better sample efficiency?
>
> **A4**:  Our ablation study (Figure 7 and Figure 8) demonstrates how the sampling and regularization components contribute to the improved sample efficiency of the proposed method.
>
> - First, the annealed sampling strategy plays a critical role by balancing guidance from the LLM policy and the agent’s own policy during the imitation learning phase. By gradually reducing reliance on the LLM policy as training progresses, the agent is guided effectively in the early stages, avoiding poor exploratory behaviors while progressively learning to depend on its own policy. This strategy ensures a smooth transition from imitation learning to reinforcement learning, as evidenced by the superior convergence speed of our model compared to the baselines. In the ablation study, the Annealed Sampling model (III: NP+NS) achieves higher success rates and faster training compared to both the Base Model and Optimized Prompt, highlighting the importance of annealing in leveraging LLM guidance during exploration and contributing to better sample efficiency.
>
> - Second, The KL-regularization term further enhances sample efficiency by aligning the policy network and value network with the LLM policy, enabling the agent to utilize structured guidance while ensuring stable updates. This regularization is particularly important during the imitation learning phase (IV: NP+NS+IL), where incorporating KL-regularized value updates improves the training of the value network, which in turn enhances exploitation in the reinforcement learning phase. Without these value updates, as shown in the ablation study (IV), the agent requires more training steps and achieves slightly worse performance.
>
> - Last, when both the annealed sampling strategy and KL-regularization (policy and value updates) are integrated into our proposed model (V: NP+NS+IL+TV), the agent achieves the best performance among all tested variants. The smooth transition from the imitation learning phase to the reinforcement learning phase allows for effective initialization of the policy and value networks, ensuring high sample efficiency in both exploration and exploitation.
>
> We hope our clarifications have addressed your concern!

---

> > ### Comment · Reviewer_LwQs · 2024-11-28
> >
> > I appreciate the authors' clarifications and the ablation study, which effectively reflect the contributions of the two components. While I find the experiment solid and persuasive and have increased the score accordingly, I remain somewhat conservative regarding the novelty of the work.

---

### Official Review · Reviewer_QL21 · 2024-10-25

**Soundness:** 2
**Presentation:** 1
**Contribution:** 1
**Rating:** 3
**Confidence:** 5

**Summary:**

This paper introduces a framework IHAC, which combines hierarchical reinforcement learning with large language models to solve complex and sparse-reward environments, where high-level actions, e.g., macro actions or skills are provided. In phrase I, IHAC first leverages LLMs to sample heuristic actions. It applies an annealing strategy to decrease the reliance on LLM progressively. while training, RL agents learned the policy and value in an imitation style. In phase II, it directly uses the standard RL algorithm to train with the learned policy and value function. Empirical studies show that IHAC outperforms baseline methods on MiniGrid, NetHack and Crafter, in terms of sample efficiency and success rate.

**Strengths:**

Empirical studies show that IHAC outperforms baseline methods on MiniGrid, NetHack and Crafter, in terms of sample efficiency and success rate.

**Weaknesses:**

This paper suffers from several critical weaknesses:

1. This work has nothing to do with hierarchical RL, however, this concept seems to be the key point and contribution of the paper. Hierarchical RL usually learns both high-level planning and low-level control. However, in this work, high-level actions are already pre-defined and provided and the agent does not learn the low-level control. The setting degenerates into the most common single-layer RL, just like the common robotics setting where high-level skills are provided. Lines 172-180 also do not show the mapping from high-level action to low-level control.

2. The proposed algorithm is trivial and theoretically incorrect. In phase I, the learned value can only be applied to offline policy, since the agents also use LLM to sample actions. However, in Line 201, the authors claimed to run a standard RL algorithm like PPO. Moreover, PPO learns V(s) instead of Q(s, a), which is also incompatible with the proposed method.

3. The paper is poorly written. Despite the incorrectly used concept of hierarchical RL, which is extremely confusing, the authors have a very limited study of works that leverage LLM to facilitate RL training. HRL and Imitation learning are not necessary to be mentioned in the related work and the LLM Agent section is quite unrelated to the topic of this paper. All the equations lack detailed explanation and analysis.

4. Due to the limited related work study, the baseline selection is also quite limited. Related work, which is not limited to GLAM[1], TWOSOME[2], SayCan[3], DigiRL[4] and ArCHer[5], are not discussed and compared.

[1] Carta, Thomas, et al. "Grounding large language models in interactive environments with online reinforcement learning." International Conference on Machine Learning. PMLR, 2023.

[2] Tan, Weihao, et al. "True Knowledge Comes from Practice: Aligning Large Language Models with Embodied Environments via Reinforcement Learning." The Twelfth International Conference on Learning Representations.

[3] Brohan, Anthony, et al. "Do as i can, not as i say: Grounding language in robotic affordances." Conference on robot learning. PMLR, 2023.

[4] Bai, Hao, et al. "Digirl: Training in-the-wild device-control agents with autonomous reinforcement learning." arXiv preprint arXiv:2406.11896 (2024).

[5] Zhou, Yifei, et al. "Archer: Training language model agents via hierarchical multi-turn rl." arXiv preprint arXiv:2402.19446 (2024).


Other issues:
1. Algorithm 1 should have a caption.

2. Line 239 see XXX is not replaced.

3. Line 274 typo: importantbt
4. Equation in Line 248 does not have a label.

**Questions:**

1. How does LLM calculate the probability of high-level action in the equation shown in Line 248?

2. What is the second KL divergence used for in equation 1? And why the first KL divergence is inversed compared to the second one?

3. What does it mean in Lines 318-319? "For all baselines, we did not train them until they converged".

---

> ### Author Response · Authors · 2024-11-25
>
> Thank you for your comments.
>
> **Q1**: This work has nothing to do with hierarchical RL, however, this concept seems to be the key point and contribution of the paper. Hierarchical RL usually learns both high-level planning and low-level control. However, in this work, high-level actions are already pre-defined and provided and the agent does not learn the low-level control. The setting degenerates into the most common single-layer RL, just like the common robotics setting where high-level skills are provided. Lines 172-180 also do not show the mapping from high-level action to low-level control.
>
> **A1**:
> Thank you for your comment. We acknowledge that our work does not involve learning low-level actions, as this is not a necessary component of our approach. However, we believe that the option-based framework in hierarchical reinforcement learning (HRL) does not inherently require learning the option space. As established in prior works, such as Sutton et al. (1999) [1], the distinction between hierarchical RL and standard RL primarily lies in the use of temporal abstractions, such as options, instead of primitive actions. In our work, we explicitly utilize time-dependent options for decision-making, which aligns with this key characteristic of HRL.
>
> It is also important to clarify that hierarchical RL is not the central contribution of our work. Instead, our key contributions are:
> - First, we introduce a novel framework that leverages large language models (LLMs) to guide high-level decision-making, particularly during the early stages of training. Our framework, IHAC, uses an external LLM to determine high-level options. This approach harnesses the LLM’s ability to provide actionable guidance, which is especially valuable when the agent lacks sufficient experience with the environment. As training progresses, our framework transitions to a standard RL algorithm to refine the policy, achieving a balance between LLM guidance and computational efficiency.
> - Second, we propose an Adaptive Sampling Strategy that combines inputs from both the RL agent and the LLM during the imitation learning phase, resulting in more effective action derivation.
> - Additionally, we design a high-level policy action distribution and a corresponding high-level value function to effectively guide learning. These innovations accelerate RL training by seamlessly combining imitation learning and reinforcement learning phases.
>
> We believe these contributions demonstrate the value of our approach, even though it does not involve learning low-level control. If you have additional questions or require further clarification, we would be happy to address them.
>
> [1] Sutton, R. S., Precup, D., & Singh, S. (1999). Between MDPs and semi-MDPs: A framework for temporal abstraction in reinforcement learning. Artificial Intelligence, 112(1-2), 181-211.
>
>
>
> **Q2**:  The proposed algorithm is trivial and theoretically incorrect. In phase I, the learned value can only be applied to offline policy, since the agents also use LLM to sample actions. However, in Line 201, the authors claimed to run a standard RL algorithm like PPO.
>
> **A2**: We believe there may be a misunderstanding regarding the design of our algorithm. Specifically, in Phase I, IHAC learns a value function while actions are sampled through the LLM. In Phase II, IHAC transitions to running PPO using the advantage function, which is based on the value function learned during Phase I. This ensures that all steps in our algorithm are theoretically sound and consistent with reinforcement learning principles.
>
> We also respectfully disagree with the characterization of our algorithm as "trivial." IHAC incorporates several novel design elements.
> First, it uses separate learning phases to effectively leverage LLM guidance during the initial stage and refine the policy through reinforcement learning in the later stage. Second, it employs an adaptive sampling strategy that dynamically combines inputs from both the LLM and the RL agent during imitation learning. Third, it introduces KL-regularized terms applied to both the policy and value functions, which enhance stability and efficiency during training.
>
> These contributions represent significant advancements over existing methods and have not been explored in prior work. We hope this clarification addresses your concerns regarding both the theoretical soundness and the innovative aspects of our approach. Please feel free to reach out if further clarification is needed.

---

> > ### Author Response · Authors · 2024-11-25
> >
> > **Q3**: PPO learns V(s) instead of Q(s, a), which is also incompatible with the proposed method.
> >
> >
> > **A3**: We believe there may be a misunderstanding about how PPO operates. According to OpenAI’s documentation and the original PPO paper, the algorithm relies on the advantage function $ A(s, a) $, which is defined as: $A(s, a) = Q(s, a) - V(s)$ This indicates that while PPO explicitly learns $ V(s) $, it implicitly involves $Q(s, a)$ through its relationship with $A(s, a)$. The advantage function $A(s, a)$ is a critical component of PPO, as it is used to optimize the policy. In our proposed method, this relationship is fully preserved. During Phase I, the imitation learning process updates both the policy $\pi_\theta $ and the value network $V(s)$, ensuring that the subsequent reinforcement learning in Phase II is well-supported. This approach aligns seamlessly with PPO’s advantage-based optimization framework and its indirect reliance on $ Q(s, a)$. We hope this clarification resolves your concern. Please feel free to reach out if further explanation is needed.
> >
> >
> > **Q4**: The paper is poorly written. Despite the incorrectly used concept of hierarchical RL, which is extremely confusing, the authors have a very limited study of works that leverage LLM to facilitate RL training. HRL and Imitation learning are not necessary to be mentioned in the related work and the LLM Agent section is quite unrelated to the topic of this paper. All the equations lack detailed explanation and analysis.
> >
> >
> > **A4**:
> > We respectfully disagree with your point and would like to provide further clarification. The key contribution of our work lies in the integration of LLM-based imitation learning, making concepts such as hierarchical reinforcement learning (HRL) and imitation learning directly relevant to our research. Specifically, HRL provides the theoretical basis for our option-based framework, which leverages high-level actions to efficiently guide decision-making, while imitation learning plays a crucial role in our first phase by utilizing LLM-generated guidance to accelerate policy optimization.
> >
> > Furthermore, we reference LLM agents because our work aligns with the broader goal of developing LLM-powered agents capable of addressing complex, language-based problems. This directly connects to our two-phase design, where the LLM provides high-level decision-making support in the imitation learning phase. As such, we believe the inclusion of LLM agent-related discussions is relevant and essential to the context of our work.
> >
> > Regarding the equations, we are happy to provide more detailed explanations if necessary. Could you please point out specific examples where you found the explanations lacking or unclear? We would gladly expand on those points in the paper to ensure all aspects of the methodology are clearly communicated.
> >
> >
> > **Q5**: Due to the limited related work study, the baseline selection is also quite limited. Related work, which is not limited to GLAM[1], TWOSOME[2], SayCan[3], DigiRL[4] and ArCHer[5], are not discussed and compared.
> >
> > **A5**: Thank you for highlighting these works. After carefully reviewing the mentioned references, we note that all the works [1-5] focus on fine-tuning existing LLMs for various tasks. In contrast, our method does not involve fine-tuning LLMs. Instead, we utilize LLMs as assistants to guide and fine-tune our reinforcement learning policy, which is represented by a much smaller policy network compared to an LLM. As a result, we did not include these works in our comparisons, as their methodologies and objectives differ significantly from ours, making direct comparisons less relevant. We hope this clarifies our baseline selection and approach.
> >
> >
> > **Q6**: How does LLM calculate the probability of high-level action in the equation shown in Line 248?
> >
> > **A6**: The high-level action distribution in our framework is represented as a one-hot vector. Specifically, for a given state and option, the action provided by the ActionNet is fixed and deterministic. Instead of computing a probabilistic distribution, we use a one-hot encoding that assigns a probability of 1 to the most likely action (as determined by ActionNet) and 0 to all other actions. This one-hot vector serves as the high-level action distribution for subsequent processing in our framework. By directly using the one-hot representation, we avoid the need for additional computation of a full probabilistic distribution while maintaining consistency in selecting the most appropriate high-level action.

---

> > > ### Author Response · Authors · 2024-11-25
> > >
> > > **Q7**: What is the second KL divergence used for in equation 1? And why the first KL divergence is inversed compared to the second one?
> > >
> > > **A7**:  The second KL divergence term aims to bound the difference between value functions. It suggests a fixed version of our policy, similar to the concept of a target value network in DQN. This term helps stabilize the value update by ensuring consistency between the learned value function and the fixed policy derived from $\pi_{\text{LLM}}$​. By doing so, it reduces potential oscillations or instability that may arise during value updates, ensuring that the value network better approximates the expected returns guided by the LLM policy. Regarding the first KL divergence term, we acknowledge that its direction is incorrectly stated in the original Equation 1 due to a typo. This has been corrected in the revised version of the paper to ensure the proper formulation of the loss function.
> > >
> > > **Q8**: What does it mean in Lines 318-319? "For all baselines, we did not train them until they converged".
> > >
> > > **A8**: To ensure a fair comparison with other LLM-assisted models, we standardized the training process by terminating all models at the same iteration count, even if some baselines had not yet converged. We have revised the wording to make this point clearer.

---

> ### Comment · Reviewer_QL21 · 2024-11-30
>
> Thanks for the clarifications, which solve some concerns. However, my major concerns still remain:
>
> 1. I am pretty sure that this work is not HRL. I am glad to see that the authors mentioned the sMDP paper. This submission actually works in an sMDP setting with temporally-extended actions, which indeed has some overlays with HRL but not exactly the same. None-HRL methods, like PPO can be directly applied to this setting without any modifications. And it is also a common setting in robotics where the low-level actions are pre-defined skills and the high-level policy is learned by RL. Readers will be misled by the term, "Hierarchical", and assume the low-level actions will also be updated. The references I provided are also in this setting. None of them claims that they are HRL.
>
> 2. It seems that the authors are not familiar with the LLM+RL research, which also partially caused the previous issue. In the related work section, works like ReAct and  Reflexion have nothing to do with this setting and should not be mentioned here. The works I provideded studied exactly the same setting, however, it is a pity that the authors refused to even mention them since they use RL to train LLM instead of a small network, even though the subsection is LLM agent.
>
> 3. I agree that using a pre-trained value function can somehow empirically helps PPO learn faster, however, it definitely has some theoretical issues. PPO's actor is initialized from scratch. The sampled trajectory has a mismatch with the pre-trained critic. I recommand authors to provide more analysis why your algorithm can bridge this gap.

---

> > ### Author Response · Authors · 2024-12-01
> >
> > Thank you for your response. We address your concerns as follows:
> >
> > **Q1**: Readers will be misled by the term, "Hierarchical", and assume the low-level actions will also be updated.
> >
> > **A1**: Regarding your concern about the concept of "hierarchical reinforcement learning (HRL)," we believe this might stem from differing interpretations across communities. Some earlier literature on HRL, such as [1], suggests that the option framework should be considered an approach to HRL, regardless of whether the options or sub-goals are learned or predefined (as in our case). The concept of HRL has been studied for over two decades, and we believe it is important to reference its original definition rather than later works that may have introduced variations or misinterpretations of the term.
> >
> > [1] Barto, A. G., & Mahadevan, S. (2003). Recent advances in hierarchical reinforcement learning. Discrete Event Dynamic Systems, 13, 341-379.
> >
> > **Q2**: Works like ReAct and Reflexion have nothing to do with this setting and should not be mentioned here. The authors refused to mention works about using RL to train LLM instead of a small network, even though the subsection is LLM agent.
> >
> > **A2**: We agree that the works you mentioned are indeed about LLM + RL or LLM agents. However, our focus is on the setting where LLMs are used to assist the decision-making process, meaning we rely solely on an existing LLM without fine-tuning it. In this regard, both ReAct and Reflexion align with our approach, as they require only access to an existing LLM or its APIs. In contrast, the other works you mentioned involve fine-tuning the LLM itself, which is fundamentally different from our approach.
> >
> > **Q3**: I agree that using a pre-trained value function can somehow empirically helps PPO learn faster, however, it definitely has some theoretical issues. PPO's actor is initialized from scratch. The sampled trajectory has a mismatch with the pre-trained critic. I recommand authors to provide more analysis why your algorithm can bridge this gap.
> >
> > **A3**:  We appreciate your acknowledgment that we use the pre-trained value function to accelerate PPO, as this is a key point of our work. However, we would like to clarify a misunderstanding about how our algorithm learns the critic function. The critic function is indeed learned from scratch. As shown in our Algorithm 1 and Equation (1), which describe the learning process of the critic $Q_w$, we do not introduce any additional regularization terms for $Q_w$ in either Phase I or Phase II, and the critic function is initialized just as the vanilla PPO as we have said in Algorithm 1, Phase I. The objective of $Q_w$ is always to learn the critic function associated with our policy $\pi_\theta$. Therefore, we believe it is inaccurate to characterize our critic function as "mismatched".
> >
> > Regarding theoretical analysis, we acknowledge its importance but have chosen not to include it in the current work as it is beyond our current scope. A high-level approach would involve analyzing the regularized policy optimization framework, as in Zhang et al. (2024), with a refined analysis of our carefully designed LLM-assisted policy $\pi_{\text{LLM}}$ as a prior. We plan to explore this further in future work.
> >
> > Thank you again for your thoughtful comments, and we hope our responses address your concerns.

---

> > > ### Comment · Reviewer_QL21 · 2024-12-02
> > >
> > > Thanks for the further clarifications, however, none of them convinces me and solves my concerns. So I choose to keep my score as it is. I urge authors to dive deeper into cutting-edge works in this domain instead of only sticking to classical concepts.

---

### Official Review · Reviewer_hv47 · 2024-10-31

**Soundness:** 2
**Presentation:** 3
**Contribution:** 2
**Rating:** 5
**Confidence:** 3

**Summary:**

This paper proposes a new training scheme to utilize LLMs to guide RL in tasks with sparse rewards. The evaluation results show consistent improvement over recent related works both in performance and token efficiency.

**Strengths:**

It's an interesting and novel idea to convert the common knowledge of LLMs into options to guide the policy learning process in tasks with sparse rewards, which is certainty an important research question for RL. The authors also provide a wide range of evaluation results.

**Weaknesses:**

(a) The technical contribution is a little bit of limited, that is, introducing a KL-regularized pre-training phase to typical RL training processes.

(b) Extra hyperparameters are introduced, such as \alpha, \lambda_t, and the number of training iterations of Phase 1.

(c) From the ablation study results, it seems that PPO only already performed well enough. It's important to show that the new algorithm can perform significantly better in scenarios where PPO only would fail.

**Questions:**

Please see the weakness part.

---

> ### Author Response · Authors · 2024-11-25
>
> We would like to thank you for your thoughtful feedback and valuable comments on our work.
>
> **Q1**: Novelty is limited as it primarily introduces a KL-regularized pre-training phase to the standard RL training process.
>
> **A1**:
> Thank you for your feedback. We would like to clarify that our primary contribution extends beyond the introduction of a KL-regularized imitation learning phase. Specifically, we believe our approach is novel in the following ways:
>
> First, our algorithm is designed to fully utilize the LLM during the early stages of training, providing effective guidance when the RL agent has limited experience with the environment. This significantly accelerates the initial learning process. Importantly, as training progresses, our method transitions to relying on the agent’s own policy and value network, enabling it to operate independently of the LLM in the later stages. This lightweight design makes our approach more practical and computationally efficient compared to other LLM-based RL methods, which often depend on the LLM throughout the entire training process.
> By reducing reliance on the LLM during the later training phases, our approach minimizes token usage, making it cost-effective and scalable—particularly in environments requiring prolonged interactions or extensive exploration.
>
>
> Second, we propose a novel mechanism that balances guidance from the high-level policy distribution and the agent’s policy using an adaptive imitation ratio. This ensures a smooth transition from exploration to exploitation (as detailed in the "Annealing Strategy in Sampling" section of the paper). This mechanism enables a gradual and controlled shift from reliance on external guidance to the agent’s autonomous decision-making, enhancing overall training efficiency.
>
> Finally, unlike existing methods that focus solely on policy updates, we introduce KL-regularization terms for both the policy and value networks. Regularizing both components ensures that the value network effectively contributes during the later training phases, resulting in improved efficiency and performance (see Equation (1) in the paper). Our ablation studies (Section 4.5, Figure 6) demonstrate the significance of this design choice. Training the policy alone without updating the value network leads to significantly lower efficiency. While performance is similar during the imitation learning phase, the policy-only method improves much more slowly during the reinforcement learning phase, requiring additional iterations for the value network to converge through PPO. This highlights the critical importance of training a high-quality value network in our method.
>
>
> **Q2**: Extra hyperparameters are introduced, such as \alpha, \lambda_t, and the number of iterations of Phase 1.
>
>
> **A2**:
> Thank you for pointing out the introduction of additional hyperparameters such as $\alpha, \lambda_t$, and $p$ (the number of iterations in Phase 1). To address this concern, we have conducted a detailed sensitivity analysis, which is included in the appendix of the revised paper. The sensitivity analysis demonstrates that while these hyperparameters are indeed introduced, their impact on the overall performance is minimal, as long as they are chosen within reasonable ranges. Specifically,
>
> - For $\alpha$, our analysis shows that the model remains robust across different values of $\alpha$, with no significant degradation in performance as long as the balance between policy and value updates is maintained.
>
> - For $\lambda_t$, our results reveal that $\lambda_t$​ primarily serves to gradually transition the agent from LLM guidance to autonomous policy learning. As shown in our analysis, the performance is largely unaffected by variations in $\lambda_t$​, provided that the decay schedule ensures a smooth reduction in LLM influence.
>
> - The number of iterations in Phase 1 determines the duration of LLM-guided imitation learning. Our results indicate that the performance is relatively insensitive to changes in this parameter, as long as Phase 1 provides sufficient guidance for the agent to initialize its policy effectively.
>
> We hope this clarification, along with the added sensitivity analysis, addresses your concern. Thank you for helping us improve the clarity and completeness of our work.

---

> > ### Author Response · Authors · 2024-11-25
> >
> > **Q3**: From the ablation study results, it seems that PPO only already performed well enough. It's important to show that the new algorithm can perform significantly better in scenarios where PPO only would fail.
> >
> >
> > **A3**:
> > Thank you for raising this concern. We assume the PPO-only method you mentioned corresponds to the Base Model shown in Figure 6. As the figure indicates, the Base Model (orange curve) significantly underperforms compared to all other methods, both in terms of early-stage performance and convergence speed. This demonstrates that PPO alone (Base Model) is far from performing well enough when compared to our proposed method. Scenarios where PPO fails have been explicitly addressed in Section 4.4 (Environment Adaptation) of the paper. For example, in the Crafter environment (referenced from Hafner, 2022), PPO-only approaches fail to achieve satisfactory results. The success rates reported in Table A.1 of Hafner (2022) are as follows: Collect Wood at 83 %, Place Table at 66 %, Make Wooden Pickaxe at 21 %, and Collect Stone and Make Stone Pickaxe are nearly impossible.
> >
> > These results show that PPO alone struggles significantly with complex tasks. In contrast, our proposed algorithm achieves much higher success rates in the same environment. Specifically, our success rates are: Collect Wood at 96 %, Place Table at 95 %, Make Wooden Pickaxe at 83 %, Collect Stone at 67 %, and Make Stone Pickaxe at 14 %. These improvements highlight the importance of the imitation learning phase in handling challenging scenarios. By combining imitation learning with reinforcement learning, our method effectively addresses the limitations of PPO in environments requiring complex reasoning or long-term planning.
> >
> > Thank you again for your feedback, as it allowed us to clarify these key points.

---

> > > ### Comment · Reviewer_hv47 · 2024-12-01
> > >
> > > Thank you for the clarification. However, I still believe that the algorithmic contributions of this paper are not substantial enough for acceptance. The points outlined in A1 seem to focus more on engineering designs rather than novel methodologies. While the integration of LLMs with RL is certainly intriguing, the chosen benchmarking tasks, such as MiniGrid, are relatively simple and may not fully demonstrate the potential of the approach.
> > >
> > > I have raised my rating to 5.

---

### Official Review · Reviewer_3Ukv · 2024-11-04

**Soundness:** 2
**Presentation:** 3
**Contribution:** 2
**Rating:** 5
**Confidence:** 3

**Summary:**

The paper proposes a two-stage algorithm aimed at enhancing exploration and reducing token usage. In the first stage, the RL agent imitates the policy generated by the LLM to improve its exploration capabilities. This phase leverages the high-level guidance from the LLM, helping the agent navigate the environment more efficiently. In the second stage, a vanilla PPO approach is applied to fine-tune the policy. The paper compares this two-stage algorithm with existing methods like LLMxHRL and LLM4Teach across several environments, including Minigrid, NetHack, and Crafter. Results indicate that this approach not only achieves higher performance but also significantly lowers token consumption, demonstrating both efficacy and efficiency in LLM-guided RL tasks.

**Strengths:**

1. This research addresses a key problem in LLM agents, using LLMs as intrinsic reward generators to enhance RL algorithm's sample efficiency.

2. The proposed algorithm is straightforward, with clear writing and experiments supporting the method's claims.

3. Experimental results on Minigrid, NetHack, and Crafter show that the method outperforms LLMxHRL and LLM4Teach in performance and token efficiency.

**Weaknesses:**

1. The reduction in token use may be due to early stopping: Table 3 shows a 10% pre-training percentage, and Figure 8’s last row indicates that success rates for 3 of 4 tasks saturate after 10% of training, suggesting the two-stage algorithm may reduce to a one stage algorithm. Additionally, the paper’s claim of a 90-95% reduction in token use likely stems from the fact that LLMs only generate guidance in the first phase. If the algorithm doesn't improve in the second phase, the improvement in token efficiency is less compelling. I recommend plotting the training curve and marking the transition to the second phase on the curve to highlight the effectiveness of the two-stage method.

2. ActionNet may introduce an unfair comparison: The paper uses ActionNet to translate options into low-level actions via a pre-defined mapping. It’s unclear if baseline algorithms also use ActionNet; if they don’t, this could create an unfair advantage, which should be addressed in the experimental section.

**Questions:**

I have outlined all my concerns in the weaknesses section.

---

> ### Author Response · Authors · 2024-11-25
>
> We would like to thank you for your thoughtful feedback and valuable comments on our work. We answer your questions as follows.
>
> **Q1**: The reduction in token use may be due to early stopping: Additionally, the paper’s claim of a 90-95% reduction in token use likely stems from the fact that LLMs only generate guidance in the first phase.
>
>
>
>
>
>
> **A1**: Since our algorithm consists of two phases, and LLM only plays in the first imitation learning phase, it is correct that the reduction in tokens comes from the reduction of the use of LLM. In fact, we believe that this reflects the true strength of our proposed algorithms: our proposed new frameworks can achieve the same or better performance compared with existing baselines, with much less tokens.
>
>
> **Q2**: If the algorithm doesn't improve in the second phase, the improvement in token efficiency is less compelling. I recommend plotting the training curve and marking the transition to the second phase on the curve to highlight the effectiveness of the two-stage metho
>
> **A2**: We appreciate your suggestion regarding the importance of reinforcement learning in the second phase. In response, we have updated Figure 8 in the revised manuscript to make the two phases more distinct for the experiments conducted on MiniGrid. Dashed lines now clearly mark the transition between the imitation learning phase and the reinforcement learning phase. As shown in Table 3, imitation learning accounts for 10% of the total training iterations, corresponding to 1.5k steps out of the 15k total steps. The updated figures provide a clearer illustration of the role and effectiveness of the second phase.
>
> Furthermore, you observed that success rates for 3 out of 4 tasks appear to saturate after 10% of training, particularly in simpler environments like KeyInBox and TwoDoorKey, where imitation learning alone suffices for success. While we agree with this observation for these simpler tasks, it does not apply to more complex environments such as SimpleDoorKey and RandomBoxKey, where reinforcement learning is essential.
>
> For the most challenging environment, RandomBoxKey, imitation learning alone is insufficient to produce satisfactory results, making the subsequent reinforcement learning phase indispensable. Our algorithm is designed not to settle for problems that can be solved solely through imitation learning but to extend its capabilities by leveraging reinforcement learning for tasks where imitation learning falls short. Our approach is further validated in more complex environments like NetHack, where the difficulty far exceeds that of MiniGrid, and imitation learning alone yields limited success. The combined two-stage framework effectively addresses both simple and complex tasks, as demonstrated by the training curves and results across diverse environments.
>
>
>
>
> **Q3**: The reviewer is concerned that the use of ActionNet to translate high-level options into low-level actions via a pre-defined mapping might give your method an unfair advantage and points out that it is unclear whether the baseline algorithms also utilize ActionNet.
>
>
> **A3**:
> Thank you for raising this important question. We would like to clarify that ActionNet, the translator used to map high-level options to low-level actions, is employed consistently across our proposed algorithm and the two baseline algorithms. This has been explicitly stated in the main paper and further emphasized in the appendix. Specifically, for any given fixed state, all three methods rely on the same ActionNet translator to determine the corresponding high-level action distribution. The primary distinction between our method and the baselines lies in how this high-level action distribution is utilized in subsequent operations. As such, we believe the comparison is both fair and valid.
>
> Please let us know if you have further concerns or require additional clarification!

---

> > ### Comment · Reviewer_3Ukv · 2024-12-02
> >
> > Thank you for your clarification. I will maintain my score for the following reasons:
> >
> > 1. Only 1 out of 4 scenarios in MiniGrid and 1 out of 2 scenarios in Nethack demonstrate the impact of the RL phase, which is insufficient to support the proposed method's advantage.
> > 2. The scenarios chosen in the paper are toy examples, and even in Crafter, they selected an academic scenario. I encourage the authors to test on more complex environments, such as the full Crafter game, which may better highlight the impact of your method. The current experimental results in the paper do not convince me of the algorithm's effectiveness.

---

### Author Response · Authors · 2024-11-25
**Revision Summary**

We appreciate the valuable feedback provided by the reviewers, which has greatly helped us improve the quality of our work. Below, we summarize the major revisions made to the manuscript. All the major changes have been highlighted in blue.

- Updated Figures for MiniGrid Experiments: In response to Reviewer 3Ukv’s suggestion, we have updated Figure 8 to clearly mark the transition between the imitation learning phase and the reinforcement learning phase. The updated figure now includes dashed lines to indicate the boundary between the two phases, illustrating the effectiveness of both stages. This provides a clearer depiction of how the two-phase framework operates.

- Refined Discussion of Main Contributions: We have revised the Introduction and Conclusion sections to better highlight our key contributions. In particular, we emphasize the strategic use of LLMs during the imitation learning phase, the integration of hierarchical policy-value updates, and the lightweight, token-efficient design of our approach. These revisions address comments from Reviewer hv47 and Reviewer LwQs, clarifying the novel aspects of our framework.

- Added Sensitivity Analysis: A detailed sensitivity analysis has been added to the Appendix C to address concerns about the additional hyperparameters introduced in our method (e.g., \p, \lambda_t, and \alpha). The analysis demonstrates that these hyperparameters have minimal impact on performance, highlighting the robustness of our framework and reducing the burden of fine-tuning.

We believe these revisions address the key concerns raised by the reviewers and significantly strengthen the manuscript. Thank you for the opportunity to improve our work further.

---

### Comment · Area_Chair_C8Ac · 2024-11-29

Dear Reviewers,

This is a friendly reminder that the last day that reviewers can post a message to the authors is Dec. 2nd (anywhere on Earth). If you have not already, please take a close look at all reviews and author responses, and comment on whether your original rating stands.

Thanks,

AC

---

### Meta-Review · Area_Chair_C8Ac · 2024-12-20

**Metareview:**

The paper proposes a two-stage algorithm to tackle the challenges of complex, sparse-reward environments. In the first stage, the RL agent imitates the policy generated by the LLM to improve its exploration capabilities. In the second stage, a vanilla PPO approach is applied to finetune the policy.

The idea of converting common knowledge of LLMs into options to guide policy learning in sparse-reward tasks is novel. However, the paper has the following weaknesses.

- Current experiments are insufficient to support the proposed method's advantage. The authors are encouraged to test on more complex environments.

- The addition of a KL-regularized pre-training phase is an incremental change to typical RL training processes, and the technical contribution might be seen as somewhat limited.

- The ablation study results suggest that PPO already performs sufficiently well, and it would be helpful to demonstrate that the new algorithm performs significantly better in scenarios where PPO would fail.

**Additional Comments On Reviewer Discussion:**

Most reviewers are negative about this submission.

---

### Decision · Program_Chairs · 2025-01-22

Reject